# Role of miR-143 and miR-146 in Risk Evaluation of Coronary Artery Diseases in Autopsied Samples

**DOI:** 10.3390/genes14020471

**Published:** 2023-02-12

**Authors:** Jian Tie, Hiroki Takanari, Koya Ota, Takahisa Okuda

**Affiliations:** 1Department of Legal Medicine, Nihon University School of Medicine, Tokyo 1738610, Japan; 2Department of Interdisciplinary Research for Medicine and Photonics, Institute of Post-LED, Tokushima 7700814, Japan

**Keywords:** coronary artery disease, microRNA genes, single nucleotide polymorphism, autopsied findings, genotyping

## Abstract

Coronary artery disease (CAD) is a common and fatal cardiovascular disease. Among known CAD risk factors, miRNA polymorphisms, such as Has-miR-143 (*rs41291957 C>G*) and Has-miR-146a (*rs2910164 G>A)*, have emerged as important genetic markers of CAD. Despite many genetic association studies in multiple populations, no study assessing the association between CAD risk and SNPs of miR-143 and miR-146 was documented in the Japanese people. Therefore, using the TaqMan SNP assay, we investigated two SNP genotypes in 151 subjects with forensic autopsy-proven CAD. After pathological observation, we used ImageJ software to assess the degree of coronary artery atresia. Moreover, the genotypes and miRNA content of the two groups of samples with atresia <10% and >10% were analyzed. The results showed that the CC genotype of *rs2910164* was more frequent in patients with CAD than in controls, which was associated with the risk of CAD in the study population. However, Has-miR-143 *rs41291957* genotype did not show a clear correlation with the risk of CAD.

## 1. Introduction

Coronary artery disease (CAD) is a complex cardiac disorder caused by partial or complete focal ischemia of the myocardium. Although many advances in early detection and effective clinical managements of this devastating disorder have been made over years, it remains a common and fatal cardiovascular disease [1]. Patho-physiological cardiac injury due to temporary or permanent cessation of coronary blood flow is usually assessed by the patient’s clinical manifestation, ultrasonography, and angiography [2,3]. Among them, coronary angiography is the primary technique that uses an iodine-containing contrast agent and X-ray images to detect the extent of narrowing of coronary arteries caused by the development or rupture of atherosclerotic plaque. The pathogenesis of the formation of arterial plaque is a complex, multi-step, and multi-factorial process, and such a complexity renders a necessity to improve the diagnostic methods and treatment of CAD.

Recent studies have shown that various genetic variants are associated with the risk of CAD [4,5]. Mainly, single nucleotide polymorphisms (SNPs) in specific microRNAs (miRNAs) increase the risk of early-onset CAD [6,7,8]. Therefore, elucidating the disease’s molecular mechanism and genetic susceptibility is needed for the early detection of high-risk groups of CAD and the foundation of more targeted prevention and treatment in the future.

miRNAs are small non-coding RNAs (approximately 22 nucleotides) that regulate gene expression at the post-transcriptional level through translational repression or degradation of messenger RNA (mRNA), affecting several cellular processes [9]. Circulating miRNAs (circRNAs) have been proposed as sensitive and informative biomarkers for the diagnosis of various diseases [10,11]. Specifically, there were unique miRNA expression patterns in the plasma samples of CAD patients in which hundreds of miRNAs were detected exhibiting differential expression compared with controls [12]. In addition, miRNAs were found to be positively correlated with the metabolism of total cholesterol, LDL cholesterol, and apolipoprotein B [13], suggesting that miRNA could be potentially involved in atherosclerosis.

Overall, more than 2000 miRNAs have been identified in the human body so far, and approximately 30% of human protein-coding genes are regulated by miRNAs [14]. miRNAs are involved in various essential biological processes, such as cell proliferation, differentiation, and apoptosis [15], and regulate a variety of cellular metabolic processes, participating in vascular dysfunction, ischemic angiogenesis, and vascular restenosis by regulating gene expression [16,17]. Therefore, miRNAs are of great value as specific biomarkers for assessing myocardial infarction, atherosclerosis, CAD, heart failure and atrial fibrillation and fibrosis. Furthermore, data from multiple studies also support the role of miRNAs in coronary restenosis [18]. In addition to well-studied protein signaling pathways in cardiac biology, many ncRNAs play critical roles in normal cardiac physiology and are essential for proper development and homeostasis of heart [19]. Several miRNAs and ncRNAs such as LIPCAR have been identified as potential biomarkers associated with the etiology of heart disease. Furthermore, cardiac regulation, such as vascularization (e.g., after myocardial infarction), is highly dependent on functional ncRNAs, including circRNAs, such as cZNF292 [20]. In addition, ncRNAs participate in pathological cardiac remodeling, assuming a critical role of ncRNA as regulatory mechanisms in cardiac tissue [21]. Hypertrophic cardiomyopathy (HCM), for example, is the most common genetically linked heart disease, often coupling with some fatal complications. Today, the traditional view of the monogenic origin of HCM is being replaced by the view that it is an oligogenic disease, the clinical phenotype of which depends not only on mutations in the gene encoding the sarcomere protein in cardiomyocytes, but also on other genes. For example, miRNAs may be involved in the pathogenesis of the disease [22].

miRNAs are emerging as important post-transcriptional regulators modulating a range of molecular signaling pathways and pathophysiological cellular processes. CircRNAs in blood are now considered to be a new class of disease biomarkers that could possess great value in clinical diagnosis and provide new avenues for personalized therapy. Recent insights into the potential role of miRNAs both as therapeutic targets exerting their significant impacts on pathophysiological processes in atherosclerosis as well as valuable clinical biomarkers that may reflect disease severity underscore the potential therapeutic and diagnostic implications of miRNAs [23].

The heart circulates blood throughout the entire body by supplying tissues and organs with essential oxygen and enriched nutrients. Normal hemodynamics of the heart will be required to maintain functional cardiovascular circulation. Irregular cardiac activity can severely disrupt blood supply, leading to myocardial ischemia and sudden cardiac death (SCD) [24]. Coronary artery disease is one of the most common causes of SCD and is strongly implicated in SCD cases with a high incidence, especially myocardial infarction (MI) [25]. Acute myocardial infarction (AMI) is one of the specific causes of SCD and usually leads to death within a short period of time upon onset of heart attack. Unfortunately, pathological findings of autopsy performed on those cases presumably died of AMI are often insignificant with attempt to determine the exact cause of the death, rendering it difficult to diagnose accurately with current histopathological techniques. Furthermore, although left ventricular systolic dysfunction and the severity of heart failure symptoms are predictors of SCD, a considerable number of SCD events do not have a previous history of reduced ejection fraction or a clinical history of heart failure, and some SCD cases were diagnosed even in the absence of positive evidence following whole body autopsy and histological examinations [26,27]. Although it is usually presumed that most of these cases died of AMI, no apparent pathological changes related to AMI could be identified. Given the proven values of biomarkers in early detection and prevention in many other diseases, it is imperative to search for sensitive biogenetic markers that can be used to detect SCD, especially in those SCD found in negative autopsy.

miRNAs are small ncRNAs that functionally regulate gene expression at the post-transcriptional level mainly by binding to complementary 3′-untranslated regions to repress messenger RNA (mRNA) translation [22]. Recent studies have suggested a potential use of miRNAs as biomarkers of cardiovascular disease as miRNAs are strongly upregulated in pathological stress and various diseases [28]. Moreover, assessing whether specific cardiac miRNAs are overexpressed in heart samples from subjects who died of AMI or SCD, and whether the miRNAs could help distinguish them, has become one of the hotly pursued research areas. It is generally accepted that miRNAs can play important roles in many biological processes, especially in those pathological processes leading to AMI, hypertension, atherosclerosis, heart failure and arrhythmia [29,30]. These miRNAs show high accuracy in distinguishing SCD from AMI as well as AMI from the normal population, offering benefits as biomarkers in differential diagnosis for clinicians and pathologists.

In forensic anatomy, cardiac death with poorly defined pathological features is not uncommon and represents a big challenge to pathologists attempting to gauge the cause of death. With the steady and inspiring advances in RNA research in recent years, an assessment of role of these biomarkers in determining cause of the death related to heart disorders has becomes the focus of forensic pathology, and forensic pathologists should tackle this challenging task with different and novel methods. In the field of forensic science, attention has been paid to the identification of miRNA in human body fluids and tissue organs, and drug detection [31].

It has been known that SNPs in miRNA genes affect their function. Thus, they have received extensive attention from researchers as new opportunities for forensic autopsy to determine the cause of death. In this study, we first performed systematic evaluations and pathological examinations of CAD specimens from forensic anatomical cases by classifying them as CAD or non-CAD samples. We then assessed whether the two susceptibility risk biomarker genes were associated with increased incidence of CAD. Finally, we examined the association of *rs2910164* (C>G) in the miR-146a gene and *rs41291957* (G>A) in the miR-143 gene with CAD risk to determine their effects on vascular circulating miRNA levels.

## 2. Materials and Methods

### 2.1. Study Subjects

We used 151 anatomical cases for research samples in our laboratory. We collected 5 mL of heart or peripheral venous blood, centrifuged at 8000 rpm for 10 min at 4 °C, and separated plasma and blood cells. Plasma was stored immediately at −80 °C and blood cells in RNase/DNase-free tubes at −20 °C until next processing. Anatomical cases included 81 males and 70 females, with an average age of 65.79 ± 13.32 years (57.96 ± 11.79 for males, 62.81 ± 10.87 for female). The examination objects were the left main coronary artery (#5), left anterior descending artery (#6), circumflex artery (#11), and right coronary artery (#2) [32]. Pathological sections were stained with Azan, and the extent of coronary atresia was graded according to anatomical and pathological examinations. Using the ImageJ software, we assessed coronary stenosis due to coronary atherosclerosis in the tissue sections under a microscope.

In this study, the collected samples were divided into two groups. The coronary arteries with an atresia rate ≥10% belonged to the coronary artery stenosis (CAD group), and those with an atresia rate less than 10% belonged to the non-coronary artery stenosis group (non-CAD group) [33]. Meanwhile, we collected cardiac or peripheral venous blood from these anatomical cases for SNP and miRNA analysis. To avoid the influence of hemolysis on RNA analysis, we collected samples within five days after death. We also collected venous blood from 11 healthy volunteers and kept it at room temperature, and plasma was extracted on days 1, 3, 5, 7, and 10, from which miRNA was quantified.

### 2.2. SNP Genotyping

According to the manufacturer’s instructions, genomic DNA was extracted from 0.2 mL blood cells using QIAamp DNA Mini Kit (Qiagen, Venlo, The Netherlands). The genotyping based on miRNA-146a C/G (*rs2910164*), and miRNA-143 G/A (*rs1291957*) was performed using 5′exonuclease TaqMan® SNP genotype assays (Applied Biosystems, Foster City, CA, USA) according to the manufacturer’s instructions. All analysis was performed using a StepOne^®^ Plus Real-Time PCR System. The TaqMan SNP genotyping reaction involved 10 µL containing 5 µL of 2×TaqPath ProAmp Master Mix (Applied Biosystems), 0.25 µL of 40× TaqMan SNP genotyping assay (including primers and probes), and 2.0 µL of genomic DNA. According to the manufacturer’s instructions, for TaqMan SNP^®^ genotyping, we used the known genotype samples confirmed by sequencing as the positive control.

### 2.3. RNA Isolation from the Plasma

Total RNA was extracted from all plasma using the mirVana PARIS and Native Protein Purification kit (Thermo Fisher) according to the manufacturer’s instructions. MicroRNA samples thus obtained were stored at −80 °C in RNase/DNase-free tubes until further processing. When total RNA was extracted from plasma, synthetic Caenorhabditis Elegans miR-39-3p (cel-miR-39-3p), which lacks sequence homology to human miRNAs, was added with a final concentration of 10 pM for normalization. Cel-miR-39-3p was incorporated into samples during RNA isolation after incubation with denaturing solution.

### 2.4. Reverse Transcription

Reverse transcription was performed using TaqMan™ MicroRNA reverse transcription kit (Thermo Fisher Scientific, Waltham, MA, USA). A total of 15 µL reaction comprised 0.15 µL of dNTP, 1 µL of multiScribe reverse transcriptase, 1.5 μL of reverse transcription buffer, 0.19 µL of RNase inhibitor, 3 µL of corresponding primers, and extracted RNA. The PCR conditions included 16 °C for 30 min and 42 °C for 30 min, followed by 85 °C for 5 min, according to the manufacturer’s instructions.

### 2.5. miRNA Expression

We amplified the cDNA using the miRNA specific TaqMan^TM^ SNP Genotyping Assay and TaqMan Fast Advanced Master Mix (Thermo Fisher Scientific). Then, the StepOne^®^ Plus real-time PCR system (Thermo Fisher Scientific Inc., Waltham, MA, USA) was used to detect the corresponding miRNAs. Amplification was performed in triplicates in a 20 µL reaction, including 10 µL of TaqMan^TM^ Fast Advanced Master Mix, 1.33 µL of cDNA, 1 µL of TaqMan^TM^ Fast Advanced miRNA Assay. An anatomical case without atherosclerosis was used as a reference sample and normalized using cel-miR-39-3p as an internal control according to the manufacturer’s recommendation for the plasma miRNA quantification in all experiments.

### 2.6. Statistical Analysis

We used SNPStats software to identify the SNP association [34]. Chi-square tests were used to assess differences in genotype and allele distributions between the two groups. Odds ratios (OR) and 95% confidence intervals (CI) were used to investigate the strength of the association and whether an SNP deviated significantly from Hardy–Weinberg equilibrium among all tested samples in the CAD and non-CAD groups. Moreover, correlation coefficients between various variables and miRNA expression in plasma were analyzed. *p* < 0.05 was considered statistically significant.

## 3. Results

### 3.1. Characteristics of the Subjects in Coronary Artery Occlusion Status

We classified the coronary artery occlusion rate under microscope images using ImageJ software and anatomical observations (Figure 1). The results showed that there was no significant difference in coronary artery occlusion rate between sex and age (*p* > 0.05). In all study samples, more than one third of subjects had a coronary artery occlusion rate less than 10% due to atherosclerosis. The coronary artery occlusion rates of 10–30%, 30–50%, and 50% or more accounted for more than half of the total (Table 1).

### 3.2. Gene Distribution of rs2910164a and rs41291957

We investigated the miR-146a C>G and miR-143 G>A polymorphisms and determined their genotypic distribution in CAD and non-CAD group (control subjects). Odds ratios (OR) were calculated from logistic regression analyses for age and sex. Control miRNA genotype frequencies were consistent with Hardy-Weinberg equilibrium (*p* = 0.28 for *rs2910164*, *p* = 0.22 for *rs41291957*). The miR-146a *rs2910164 C>G* polymorphism was significantly different between the CAD group and controls, assuming codominant (CG vs. GG, OR: 1.61, 95% CI: 0.91–3.35, *p* = 0.006), dominant (CG + CC vs. GG, OR: 1.59, 95% CI: 0.76–2.81, *p* = 0.0014), and overdominant (GC vs. GG + CC, OR: 1.12, 95% CI: 0.75–2.73, *p* = 0.049) inheritance. According to the lowest *p*-value, the dominant model was the best-fitted (*p* = 0.0014). Compared to the GG homozygotes, individuals with at least one C allele had a significantly higher risk of CAD. However, no interaction with any covariates was found for *rs41291957* (all *p*-values > 0.05). Therefore, miR-143 *rs41291957 G>A* polymorphism was not significantly different between the CAD group and controls (Table 2).

### 3.3. Changes in the miRNA Expression of the Plasma of the Control Samples Kept at Room Temperature

We collected blood samples from healthy individuals and stored them at room temperature to observe changes in RNA over time. These samples were used as a reference when we extracted plasma from autopsy samples for miRNA quantification. After five days, blood cells began to undergo hemolysis, which led to the efflux of intracellular RNA into the plasma, thereby increasing the plasma concentrations of miR-146a and miR-143 (Figure 1). Therefore, we used the samples collected within five days after death to avoid hemolysis.

### 3.4. Association between the SNPs Genotypes and miRNA Expression in CAD

We investigated the possible effects of *rs2910164* (C>G) on miR-146a and rs41291957 (G>A) on miR-143 expression levels using the TaqMan genotyping method. The results showed that only 18.8% of the CAD group in *rs2910164* carried the GG genotype, while more than 80% carried the GC genotype. Comparing the expression of miR-146a between the *rs2910164*-GG and CG genotypes in the CAD group, the relative expression of miR-146a in the CG genotype in the CAD group was significantly higher than that in the GG genotype group (*p* < 0.05) (Figure 2). Although the GA genotype was slightly higher than the GG, there were no significantly different expression levels between the two (*rs41291957* (G>A) vs. miR-143, *p* > 0.05) (Figure 3).

## 4. Discussion

Although the association between miRNA polymorphisms (SNPs) in protein-coding genes and the risk of CAD is being extensively studied, most of the samples used in these studies define CAD by clinical blood biochemical examination combined with imaging detection such as imaging. Because of this clinically defined CAD, the majority are individuals with more severe coronary atresia. For those individuals with relatively mild coronary artery stenosis, or in case the clinical detection is not very clear, a new attempt may be made to diagnose CAD from anatomy or pathology and to analyze the correlation of its miRNA genes. The mechanism of arteriosclerosis is very complicated, and the living environment, living habits, physical condition, genetics, etc. all have different degrees of potential influence. Arteriosclerosis brings various diseases and even death to the human body. To understand the mechanism of coronary artery disease from different perspectives, so as to provide better prevention, diagnosis, and treatment methods, is an urgent problem to be solved. In forensic anatomy, it is common to see individuals of different ages, sexes, and causes of death with varying degrees of arteriosclerosis. Although most of their causes of death were not directly caused by coronary artery disease, the extent to which CAD affects their causes of death is not very clear in most cases, and it is a question worth exploring. There are some cases where anatomical findings or morphological examination alone cannot give a definitive answer. The expression of miRNA in the blood and the study of the association between SNP and CAD may help us find the answer as an auxiliary diagnosis. This is why we conducted this study. Here, we collected individual samples of males and females of different ages in forensic necropsy and performed qualitative analysis of the coronary arteries of the study samples through anatomical observation and pathological diagnosis. This method can directly reflect the pathological state of CAD and classify the degree of arteriosclerosis more clearly. We performed an association study of two known polymorphic markers, miR-146a *rs2910164* and miR-143 *rs41291957*. In order to avoid the impact of various diseases on miRNA expression, the samples we selected did not contain cancer or other major diseases.

After clarifying the distribution of two SNP genotypes, we quantified the corresponding individual miRNA expression levels, and then assessed the influence of the presence of genetic variation and CAD susceptibility. This study supports the evidence that levels of miR-146a found affect susceptibility to CAD. Furthermore, we found that individuals carrying the C allele at *rs2910164 C>G* exhibited higher expression levels of mature miR-146a. The miR-146a polymorphism involves the inclusion of a G to C nucleotide substitution, resulting in the miR-146a precursor stem structure changing from a G:UG: U pair to a C: UC: U mismatch, resulting in individuals carrying the C allele becoming susceptible to CAD [35]. As researchers investigate the distribution of miR-146a in different populations, the mechanism of its occurrence will become more and more clear. Although there are some reports on miR-143 *rs41291957* polymorphism and CAD susceptibility [33,36], no positive evidence of association was obtained in this study. This result may be related to population genetics. miR-143 *rs41291957* A/G polymorphism was reported in Chinese population, but the results showed no significant association between genotype or allele frequency and CAD. The exact reason is unclear, and it is speculated that at least miR-143 *rs41291957* did not play a major role in the genetic susceptibility to coronary heart diseases in the study population [36].

In addition to CAD, *rs2910164* has been associated with the risk of several other diseases [37,38]. It was reported that miR-146a expression was higher in peripheral blood mononuclear cells (PBMCs) in the CAD group than in the non-CAD group [39]. It was also pointed out that the overexpression of miR-146a in PBMCs in patients with acute coronary syndrome may directly affect the differentiation of Th1 cells, which is of great significance to the progression of heart disease. miRNA is also related to human immunity, tumorigenesis, calcification of vascular smooth muscle, etc., which shows the complexity of miRNA and disease regulation [40]. Another study also showed that the expression of miR-146a was significantly elevated in atherosclerotic plaques, leading to atherosclerosis [41]. Therefore, increasing the expression of miR-146a may promote the occurrence and development of CAD. Furthermore, individuals carrying the *rs2910164* C allele in the miR-146a gene may have elevated expression levels of mature miR-146a, enhanced Th1 cell activation, and increased risk of CAD.

In order to more objectively evaluate the state of vascular atresia caused by coronary atherosclerosis, we used ImageJ software to classify the stained pathological specimens, so as to improve the accuracy of judging miRNA genes and CAD risk research. In this study, we classified samples with a coronary artery occlusion rate of less than 10% as the non-CAD control group and those with a coronary artery occlusion rate of 10% or greater as the suspected CAD group. Although we classified coronary atresia into four grades, considering the non-coronary atherosclerosis group would greatly affect the statistical results. Therefore, we classified the samples with coronary atresia rate equal to or higher than 10% into the suspected CAD group for statistical analysis. Although this division will affect the statistical difference to a certain extent, it will make the non-coronary atherosclerosis group closer to the normal control group. This classification differs from clinical studies because there are no blood biochemical and photographic data on antemortem samples, and all data come from anatomical and pathological testing. Based on the principle of conducting research under conditions as close as possible to normal subjects, the obtained data were analyzed. The statistical results also suggest that there is no statistical significance between the two groups in gender and age. The correlation between the two SNP groups of miRNAs on CAD shows different results, but it is necessary to increase the number of samples or change the sampling method in the next study to give the answer. This is a new method to study crown CAD and miRNA from the perspective of anatomy and pathology. We hope to provide some meaningful hints for similar miRNA–CAD association studies in the future.

## 5. Conclusions

This study used anatomical observation and pathological diagnosis to classify CAD and non-CAD individuals and explored the relationship between polymorphisms of miRNA genes and CAD. For individuals with more than 10% coronary atresia, miR-146a *rs2910164* was associated with an increased risk of CAD, whereas miR-143 *rs41291957* did not show a clear association.

## Figures and Tables

**Figure 1 genes-14-00471-f001:**
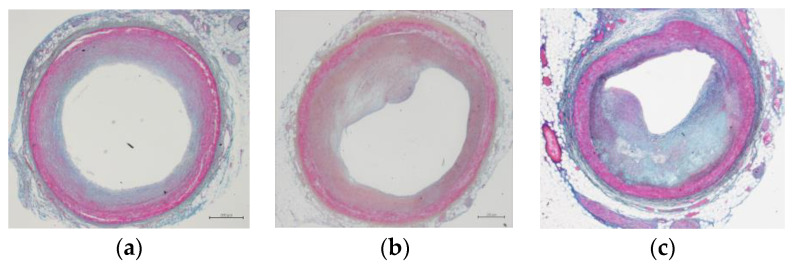
Microscopic image of coronary lumen narrowing caused by atherosclerosis after Azan staining. (**a**) shows the case of coronary artery stenosis rate below 10% (non-CAD group). (**b**,**c**) show two CAD cases with coronary stenosis rates of 35% and 77%, respectively.

**Figure 2 genes-14-00471-f002:**
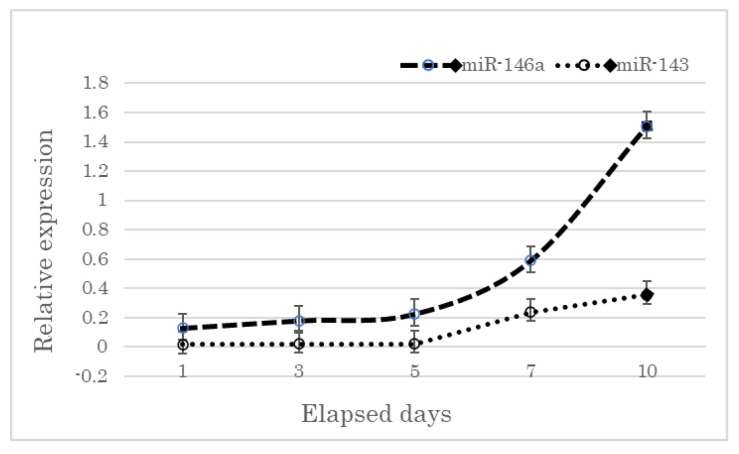
Expression levels of miR-146a and miR-143 in plasma samples of the healthy volunteers kept at room temperature for different days.

**Figure 3 genes-14-00471-f003:**
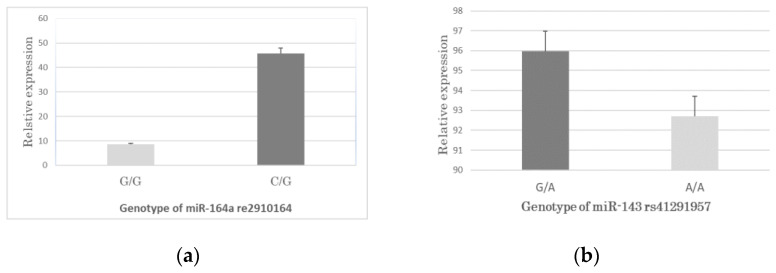
miRNA expression in the CAD group with two SNPs. (**a**) shows miR-146a with *rs2910164* G/G and C/G genotype; (**b**) shows miR-143 with *rs41291957* A/A and G/A genotype.

**Table 1 genes-14-00471-t001:** Characteristics of subjects in coronary artery occlusion status.

Closure Rate (%)	Male	Female	Total
<10	28	26	54
10–30	14	14	28
30–50	16	20	36
>50	23	10	33
Total	81	70	151

**Table 2 genes-14-00471-t002:** Baseline characteristics between CAD group and non-CAD group.

Model	Genotype	CAD Group (%)	Non-CAD Group (%)	OR (95% CI)	*p*-Value
rs2910164 C>G					
Codominant	G/G	13 (14)	2 (3.5)	1	
	C/G	60 (64.5)	16 (27.6)	1.61 (0.91–3.35)	0.006
	C/C	20 (21.5)	40 (69)	1.18 (0.82–2.31)	
Dominant	C/G-G/G	73 (78.5)	18 (31)	1	
	C/C	20 (21.5)	40 (69)	1.59 (0.76–2.81)	0.014
Recessive	G/G	13 (14)	2 (3.5)	1	
	C/C-C/G	80 (86)	56 (96.5)	0.91 (0.53–2.04)	0.32
Overdominant	C/G	60 (64.5)	16 (27.6)	1	
	C/C-G/G	33 (35.5)	42 (72.4)	1.12 (0.75–2.73)	0.0049
Log-additive					0.0025
rs41291957 G>A					
Codominant	A/A	10 (15.2)	5 (5.9)	1	
	G/A	20 (30.3)	34 (40)	0.81 (0.62–1.23)	0.26
	G/G	36 (54.5)	46 (54.1)		
Dominant	G/A-A/A	30 (45.5)	39 (45.9)	1	0.68
	G/G	36 (54.5)	46 (54.1)	0.71 (0.53–1.35)	
Recessive	A/A	10 (15.2)	5 (5.9)	1	0.17
	G/G-G/A	56 (84.8)	80 (94.1)	0.66 (0.21–1.12)	
Overdominant	G/A	20 (30.3)	34 (40)	1	0.21
	G/G-A/A	46 (69.7)	51 (60)	0.41 (0.19–0.87)	
Log-additive					0.76

## Data Availability

The data presented in this study are available on request from the corresponding author.

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
