# Peer review of "Role of miR-143 and miR-146 in Risk Evaluation of Coronary Artery Diseases in Autopsied Samples"

_genes, 2023, doi:10.3390/genes14020471_

Round 1

Reviewer 1 Report

Dear Author, 

the paper I have to review talks about the role of miRNA in CAD risk evaluation. Data shows that based on the MiRNA expression, population can be divided into two risk classes.

The text must to be justified as in word template.

In introduction, at line 43-45: "Therefore, they are considered specific biomarkers for myocardial infarction, atherosclerosis, CAD, heart failure, atrial fibrillation, and fibrosis. Moreover, several studies supported the role of miRNAs in restenosis". I think this sentence is the center of the paper, because you're going to study the role of miRNA in the CAD risk. I mean, this paper is very important in forensic science because the presented article continues the worldwide work on miRNA and the myocardial pathology. You cannot conclude the introduction without an extensive [but brief] explanation of the current knowledge of miRNA and Heart Diseases. So, please, resume you introduction at line 1-43 and better explain the involvement of miRNA in CAD. Moreover, expand your bibliography at this point with some citations, I advise you: 

10.1111/jcmm.14463

10.3390/diagnostics11010032

Line 63 please provide a statistical descriptive analysis of your population, i.e. add Age+SD  for each gender. Are there any comorbidities? Are there any negative control? 

I look forward to reading your revised paper. 

Reviewer 2 Report

I have reviewed the manuscript entitled "Role of miR-143 and miR-146 in Risk Evaluation of Coronary 1 Artery Diseases in Autopsied Samples"

Thanks to authors for this very important research.

Coronary artery disease is a very important public health problem and we need new information about its genetic origin.

This study helps to understand the genetics of coronary artery disease.

The research is well planned and the manuscript is well written.

The manuscript is suitable to be published in this form.

Author Response

Dear reviewer,

Thank you for carefully reviewed our manuscript. Your evaluation encourages us, and we are really happy.

Round 2

Reviewer 1 Report

the authors responded to the reviews